

# A review of reinforcement learning based hyper-heuristics

Cuixia Li, Xiang Wei, Jing Wang, Shuozhe Wang and Shuyan Zhang

School of Cyber Science and Engineering, Zhengzhou University, Zhengzhou, Henan, China

## ABSTRACT

The reinforcement learning based hyper-heuristics (RL-HH) is a popular trend in the field of optimization. RL-HH combines the global search ability of hyper-heuristics (HH) with the learning ability of reinforcement learning (RL). This synergy allows the agent to dynamically adjust its own strategy, leading to a gradual optimization of the solution. Existing researches have shown the effectiveness of RL-HH in solving complex real-world problems. However, a comprehensive introduction and summary of the RL-HH field is still blank. This research reviews currently existing RL-HHs and presents a general framework for RL-HHs. This article categorizes the type of algorithms into two categories: value-based reinforcement learning hyper-heuristics and policy-based reinforcement learning hyper-heuristics. Typical algorithms in each category are summarized and described in detail. Finally, the shortcomings in existing researches on RL-HH and future research directions are discussed.

## INTRODUCTION

Evolutionary algorithms (EAs) represent a category of population-based heuristics inspired by the natural selection and evolution (*Zhou et al., 2021*), which solve optimization problems by simulating mechanisms such as inheritance, mutation, and adaptive selection observed in the process of biological evolution. EAs, which are frequently applied to address challenges characterized by sample space that lack clear definition, gather evolutionary insights over generations by exploring various regions within the problem space (*Houssein et al., 2021*; *Baykasoğlu & Ozsoydan, 2017*). As they explore, EAs iteratively refine solutions until converge to a local optimum, thereby crafting tailored solutions suited to the specific problem at hand. With the capability to explore complex search spaces, EAs offer remarkable adaptability and parallelization ease (*Young et al., 2015*). At present, EAs find extensive application across diverse research domains, including sequence optimization (*Lutz et al., 2023*; *Junior et al., 2023*), scheduling (*Chen et al., 2023*; *Wang, Li & Gao, 2023*; *Wang et al., 2023*), object recognition (*Afif et al., 2023*; *Zhang, Li & Qi, 2023*).

The concept of heuristic search is introduced under the framework of EAs, paving the way for the development of hyper-heuristics (HH) (*Cowling, Kendall & Soubeiga, 2001*), which aims at enhancing the optimization process with greater intelligence. In recent years, HH has emerged as a prominent research domain, yielding notable advancements

Corresponding author
Shuyan Zhang, syzhang@zzu.edu.cn

across various fields such as traveling salesman problem (*Pandiri & Singh, 2018*; *Dasari & Singh, 2023*; *Simões, Bahiense & Figueiredo, 2023*; *Gharehchopogh, Abdollahzadeh & Arasteh, 2023*), packing problem (*Ross et al., 2002*; *Leon, Miranda & Segura, 2009*; *Guerriero & Saccomanno, 2022*; *Guerriero & Saccomanno, 2023*), examination timetabling problem (*Ahmadi et al., 2003*; *Burke et al., 2005*; *Burke, Petrovic & Qu, 2006*; *Sabar et al., 2012*), vehicle routing problem (*Hou et al., 2022b*; *Asta & Özcan, 2014*; *Hou et al., 2022a*; *Shang et al., 2022*), and scheduling problem (*Cowling, Kendall & Han, 2002*; *Cowling, Kendall & Soubeiga, 2002a*; *Cowling, Kendall & Soubeiga, 2002b*). HH, which aims to effectively solve various real-world optimization problems (*Özcan, Bilgin & Korkmaz, 2008*), combines the problem specificity of heuristic search with the global search capability of EAs. HH is designed not for the exclusive resolution of specific domain problems, but rather to steer the search process utilizing high-level strategies. Concurrently, HH excels at developing universal methodologies by distilling and generalizing problem attributes. This empowers HH with the ability to address cross-domain problems adeptly and handle diverse optimization problems with finesse. Furthermore, HH can be delineated into two distinct categories depending on the heuristic search space's characteristic, namely generation hyper-heuristics algorithm and selection hyper-heuristics algorithm. Selection hyper-heuristics dynamically and selectively utilize diverse human-designed heuristics or components based on the evolving requirements and performance metrics throughout the iterative process. Selection hyper-heuristics can also reduce manual intervention and enhance flexibility, thereby the application of selection hyper-heuristics is more widespread. The selection hyper-heuristics algorithm consist of two important modules: learning and selection. Based on what has been learned, the selection hyper-heuristics dynamically choose an appropriate low level heuristic (LLH) and apply the LLH to address the given problem. The performance and efficiency of HH hinge directly on the chosen learning strategy, which assumes a critical role. Commonly used learning methods encompass methods based on meta-heuristics algorithm, selection function, and reinforcement learning.

Reinforcement learning based hyper-heuristics (RL-HH) uses reinforcement learning (RL) (*Sutton & Barto, 1998*) as learning method. Compared with meta-heuristics and selection function, RL stands out by not demanding in-depth prior knowledge of the problem domain. RL exhibits robust generalization capabilities, allowing RL to automatically learn to adapt to the dynamic environment at runtime. For these above reasons, RL proves to be more proficient when confronted with novel challenges. The ideas of HH and RL revolve around finding the most suitable solution through specific strategies or methods. Both entail an iterative search process, optimizing their performance through continuous trial and error and feedback. However, there are essential differences between RL and HH. HH is a type of optimization algorithm that focuses on learning how to choose the appropriate LLH at the decision point. While RL is a machine learning method that relies on the reward mechanism of environmental feedback for learning. So HH and RL are based on different principles. However, because RL's focus on long-term reward maximization matches HH's goal of pursuing the optimal solution, RL can be combined with HH to aid in making the most advantageous single-step decision in the global search process. Recognized as a prevalent research topic in machine learning, RL has found broader

applications in fields such as robot control (*Kober, Bagnell & Peters, 2013*; *Mülling et al., 2013*; *Banino et al., 2018*; *Mirowski et al., 2016*), traffic control (*Belletti et al., 2018*; *Wei et al., 2018*; *Chu et al., 2020*; *Han, Wang & Leclercq, 2023*), game (*Mnih et al., 2013*; *Moravčík et al., 2017*; *Hessel et al., 2018*; *Wang et al., 2020*), optimization and scheduling (*Ipek et al., 2008*; *Liu et al., 2023a*; *Liu, Piplani & Toro, 2023*; *Tejer, Szczepanski & Tarczewski, 2024*). RL is an autonomous learning method in which an agent can learn from environmental interactions without heavy reliance on labeled data or prior knowledge. RL is capable of adapting to changing situations and needs, while also automatically generating training samples and learning in a real-time environment. RL continuously improves strategies based on feedback information to effectively handle complex and dynamic problems. Consequently, the innovative combination of HH with global search capabilities and RL with learning capabilities has sparked a burgeoning research trend in optimization, giving rise to RL-HH as an innovative frontier. The execution process of RL-HH can be described as follows: (i) Initialization is performed at the beginning of the algorithm to ensure consistent performance of all LLH; (ii) Select and apply LLH to generate new solutions, and the RL agent updates the performance of each LLH based on the quality of the solution; (iii) At each decision point, the LLH to be applied next is selected based on the performance of the known LLH; (iv) Repeat steps (ii) and (iii) until the termination condition is met.

## Rationale for the review

To the best of our knowledge, there are few studies describing the survey in the field of HH. *Özcan, Bilgin & Korkmaz (2008)* investigated and summarized HH, but the review they wrote only focused on the general classification of HH. With RL increasingly used in solving complex problems, embedding RL into HH to select the most appropriate LLH has become more and more popular. However, there has not yet been a comprehensive survey and summary of RL-HH, which motivates the need for a review of the field. This review aims to fill this need by systematically organizing and analyzing existing research, providing academia and industry with an in-depth understanding of RL-HH. By enabling more people to better understand the latest developments in this field, as well as the advantages, disadvantages and possible future research directions of the method, this review will contribute to advancing the field of hyper-heuristics.

## Who this study is intended for

Given the crucial role that excellent RL-HH plays in the field of optimization, the review is intended for a variety of audiences to provide a comprehensive understanding of the current development of RL-HH. Audiences for this review include, but are not limited to:

(1) RL and optimization researchers: For those researchers who are interested in the study of RL and HH, through this review they can gain insights into the latest progress and technology in the field of RL-HH, which will be helpful for future research work.

(2) Practitioners in the industry: Professionals engaged in related fields such as artificial intelligence, optimization or decision analysis in the industry can learn how to apply RL-HH to actual industrial problems and solve real-world optimization problems in various industrial settings.

(3) Managers in the corporate world: Managers and decision-makers in the corporate world may be interested in learning how to leverage RL-HH to improve business processes, optimize resource allocation, and increase efficiency.

## SEARCH METHODOLOGY

Regarding the theme of 'A review of reinforcement learning based hyper-heuristics', we use the following methods to retrieve and organize previous relevant research articles, aiming to better understand the development trends in this field.

Searching for literature: A systematic approach is used to search and select articles, thus ensuring comprehensive and unbiased coverage of the literatures. We select six academic digital sources, and specific search terms were applied to these repositories to identify literatures related to reinforcement learning based hyper-heuristics. The bibliographic databases considered are: Scopus; Web of Science (WoS); IEEE Xplore; SpringerLink; and ACM Digital Library. Additionally, the academic search engine Google Scholar is also used to find any documents that might have been missed to ensure a comprehensiveness.

Keywords: Selecting popular keywords from related fields to screen out literature highly related to reinforcement learning based hyper-heuristics. The keywords include: "hyper-heuristics", "selective hyper-heuristic algorithm", "optimization", "reinforcement learning", "artificial intelligence", and "deep learning". The search terms ultimately applied to all databases were ("hyper-heuristics" OR "selective hyper-heuristic algorithm" OR "optimization") AND ("reinforcement learning" OR "artificial intelligence" OR "deep learning").

Selection criteria: Carefully read the titles and abstracts of these literatures after obtaining a large list of literatures. Conduct a thorough analysis of these literatures to determine their relevance to our research field and their eligibility. Based on this, relevant articles that fit our research field are selected.

Minimize bias: Striving to minimize bias, the search and selection process endeavors to include diverse perspectives while maintaining standards of relevance and quality. Each chosen literature underwent meticulous scrutiny to assess scientific rigor, ensuring the literature review presents a fair and comprehensive overview of existing research without undue favoritism towards any specific approach or viewpoint.

## RELATED WORKS

### Hyper-heuristics

HH was proposed by *Denzinger, Fuchs & Fuchs (1997)* as early as 1997, and later supplemented and defined by *Cowling, Kendall & Soubeiga (2001)* as heuristics to choose heuristics. HH is a search method that solves various optimization problems by automatically selecting or generating a set of heuristics, thereby providing a more general search paradigm. With rapid implementation and simplicity, HH evolves through the acquisition of performance feedback from LLH during the search process (*Pylyavskyy, Kheiri & Ahmed, 2020*). The core concept of HH revolves around leveraging multiple LLHs to address the given problems, fostering a comprehensive framework to intelligently

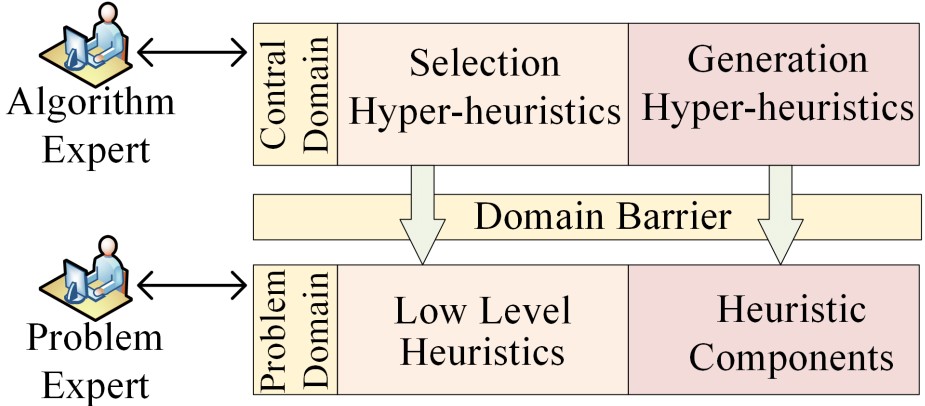

**Figure 1** The framework of hyper-heuristics.

control the applications of LLHs (*Garrido & Castro, 2012*). HH exploits the search space of LLHs and adaptively discovers heuristic strategies to address the problem rather than directly producing a solution. HH, which essentially has the ability to learn, entails gaining experience from the current running results and adjusting in a direction that is beneficial to solving the problem. As an advanced automated method for selecting or generating a set of heuristics, HH operates on an elevated plane, managing or generating LLHs that operate within the domain of the problem (*Burke et al., 2013*; *Özcan et al., 2010*). Capable of integrating efficient optimization methods to better solve complex problems, HH pursues the versatility and adaptability (*Burke et al., 2003*). HH typically consists of a dual-layered framework encompassing both a control domain and a problem domain (*Burke et al., 2013*). The problem domain involves diverse LLHs and heuristic components, while the control domain serves as a high-level strategy for choosing suitable solutions within the problem domain. A domain barrier within the problem environment is established between these two layers, which streamlines algorithmic design complexities across distinct problems and bolsters the universal applicability of HH across diverse problem scenarios. The execution framework of HH is shown in Fig. 1.

    *Burke et al. (2010)* classified HH into two categories: selection hyper-heuristics and generation hyper-heuristics. Selection hyper-heuristics make astute choices regarding the appropriate LLH to apply at each decision point from a collection of human-crafted heuristics. In contrast, generation hyper-heuristics aim to create novel LLHs by utilizing pre-existing components (*Choong, Wong & Lim, 2018*). Given that selection hyper-heuristics directly select the appropriate LLH from available options, there is no need for time-consuming searches during the solution process. HH can generate solutions in real-time, showcasing a high degree of flexibility (*Drake et al., 2020*). This research exclusively focuses on selection hyper-heuristics.

    The selection hyper-heuristics opt for a suitable LLH from a collection of LLHs to employ at the present phase (*Ferreira, Goncalves & Trinidad Ramirez Pozo, 2015*). The control domain of selection hyper-heuristics consists of two components: (i) High Level

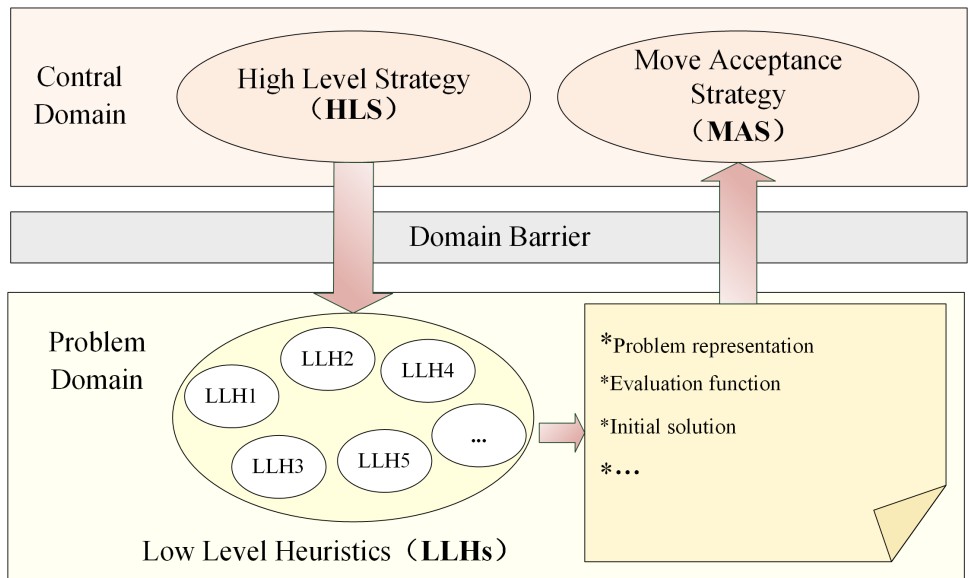

**Figure 2** Execution framework of the selection hyper-heuristics.

Strategy (HLS): HLS strategically selects which appropriate LLH to apply at each decision point from a pool of LLHs based on the learned knowledge; (ii) Move Acceptance Strategy (MAS): MAS is responsible for evaluating the acceptability of the newly generated candidate solution during the iterative process. The problem domain of selection hyper-heuristics contains a group of human-designed off-the-shelf LLHs. The selection hyper-heuristics employ HLS to select the appropriate LLH from LLHs, apply the selected LLH to tackle the problem and generate a new candidate solution. Subsequently, MAS evaluates the performance of the solution and decides whether to substitute the previous candidate solution with the new candidate solution. This iterative process continues until predefined termination criteria are met (*Alanazi & Lehre, 2016*). The execution framework of selection hyper-heuristics is shown in Fig. 2.

## Reinforcement learning in hyper-heuristics

As a learning mechanism, HH can use various methods to gather information during learning processes for LLH selection. RL is a common example which has drawn much attention as a powerful decision-making tool. Unlike traditional function selection, RL does not require intricate functions for diverse strategies, nor does it demand excessive time like meta-heuristics. Therefore, RL emerges as a powerful approach for solving optimization challenges in complex decision-making scenarios, offering distinct advantages in intelligently guiding the selection of LLH (*Choong, Wong & Lim, 2018*; *Shang, Ma & Liu, 2023*).

RL stands as a pivotal filed within the realm of machine learning that is adept at assimilating knowledge acquired from previously solved instances of a problem to inform decision-making in novel scenarios. The core concept of RL revolves around learning to

**Peer**J Computer Science

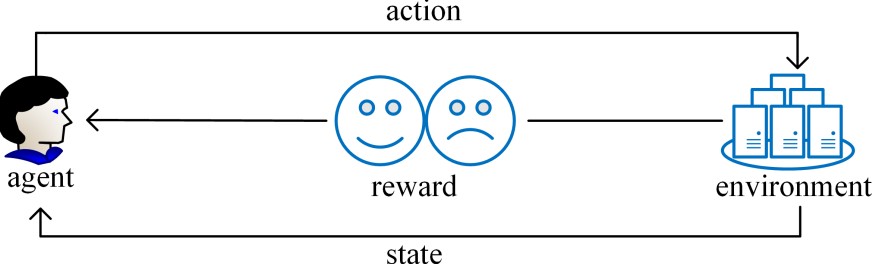

**Figure 3** **The execution process of reinforcement learning.**

evaluate actions that enabling the system to learn through trials and errors how to choose the best action in a given situation (*Kaelbling, Littman & Moore, 1996*). RL acquires knowledge through dynamic interaction and feedback between the agent and the environment with less prior knowledge. In the process of executing a certain task, the agent first interacts with the environment, and then performs action in the environment to generate a new state. At the same time, the environment will give a corresponding reward. As the cycle continues, the agent continuously adjusts learning strategy according to past experience and selects the most appropriate action. Ultimately the agent makes a series of decisions and obtains the maximum reward (*Sutton & Barto, 1998*). The goal of RL is to enable the agent to gradually improve learning strategy through continuous trial and learning feedback, enabling the agent to make increasingly insightful decisions when confronted with diverse environments and tasks. One of the characteristics of RL is its autonomy and the ability to automatically generate training data. During the training process, the agent automatically generates training samples. These samples reflect the interaction between the agent and the environment, which means that RL can be used to solve problems that are difficult to handle with traditional technologies. In addition, unaddressed problem instances yield valuable experience, which can be quantified into reward and penalty values. Guided by these values, the agent determines the next course of action. Figure 3 shows the execution process of RL.

The Bellman equation is a core mathematical tool in RL that decomposes complex multi-stage decision-making problems into a series of simple sub-problems. Bellman's equation expresses how to recursively calculate the expected return given the current state and action taken. The core idea of Bellman equation is current decisions will affect the future state, which in turn depends on subsequent decisions. The Bellman equation is shown in Eq. (1).

$$Q(s,a) = E_{s' \sim s}[r + \gamma Q(s', a')|s, a]. \tag{1}$$

When the state action value function Q is optimal, the optimal Bellman equation is obtained, as shown in Eq. (2).

$$Q(s,a) = E_{s' \sim s}[r + \gamma \max Q(s', a')|s, a]. \tag{2}$$

Through iterative solving of the Bellman equation, the agent learns how to take the best action in a given environment to maximize long-term rewards.

According to the different methods of agent action selection, RL can be categorized into two types: value-based reinforcement learning (VRL) and policy-based reinforcement learning (PRL).

### Value-based reinforcement learning

VRL centers on computing value functions to determine the appropriate action, thereby implicitly deriving a deterministic policy. Depending on whether neural networks are involved, VRL can be divided into traditional VRL (TRL) and deep VRL (DRL).

*Traditional value-based reinforcement learning (TRL)* State-Action-Reward-State-Action (SARSA) and Q-Learning stand as the two most representative methods within TRL. SARSA operates as an on-policy learning approach, where the reward value of the next action actually taken is utilized to update the value function. Different from the way SARSA updates the value, Q-Learning is an off-policy learning method that focuses on the optimal strategy during the learning process. Regardless of the action actually executed, Q-Learning consistently updates the value function based on the action that would maximize future rewards. HH focuses more on how to effectively search and optimize the solution space, rather than updating the value function in real time. The update strategy employed by Q-Learning is more helpful in guiding HH's search process to be more effective. Therefore, existing algorithms mainly use the combination of Q-Learning and HH, which is also consistent with the findings obtained through search methodology. The pseudo code of Q-Learning is as follows:

---

**Algorithm 1:** A pseudo code for Q-Learning

---

1   *Initialize $Q(s, a)$ arbitrarily;*

2   **for** *each episode* **do**

3      *Initialize state s;*

4      **while** *s is not terminal* **do**

5         Choose action *a* using policy derived from $Q$ (e.g., $\epsilon$-greedy);

6         Take action *a*, observe reward *r* and next state $s'$;

7         Update $Q(s, a) \leftarrow Q(s, a) + \alpha \left[ r + \gamma \max_{a'} Q(s', a') - Q(s, a) \right]$;

8         $s \leftarrow s'$;

9      **end**

10 **end**

---

In this pseudo code, $s$ represents the current state, $a$ represents the current action, $r$ represents the reward, $s'$ represents the next state, $a'$ represents the next action, $\alpha$ represents the learning rate. $\gamma$ is the discount rate, which is used to balance the immediate reward and future rewards.

*Deep value-based reinforcement learning (DRL)* On the basis of TRL, neural network is added to propose DRL, which fills the gap that TRL cannot solve high-dimensional tasks. One of the most representative DRL algorithm is Deep Q-Network (DQN), which enhances Q-Learning by incorporating neural networks and introducing several improvements. In addition to using a neural network to approximate the value function, DQN also uses a separate target network to generate the target Q value. Additionally, DQN utilizes experience replay mechanism during training process to reduce the correlation between samples. The addition of these modules improves data utilization, while increasing the stability and convergence speed of the learning process. Building upon DQN, Double Deep Q-Network (DDQN) further improves performance by employing two independent neural networks to estimate values and select actions respectively. DDQN alleviates the overestimation problem existing in DQN and improves learning accuracy. Duel Double Deep Q Network (D3QN) extends DDQN by integrating the dueling architecture, decomposing the value function into two parts: the state value function and the advantage function, which can learn the value of each action more effectively.

### Policy-based reinforcement learning

PRL directly parameterizes the policy and optimizes these parameters to learn a policy directly instead of learning a value function first. This type of method first initializes the strategy parameters, then executes the corresponding strategy, collects experience and calculates returns, and finally updates the strategy parameters to increase returns. The entire process is repeated until the policy performance is satisfactory or the termination condition is met.

RL represents a powerful learning paradigm that simulates the nature of human learning and continuously refines decision-making strategies continuously through trials and errors. Across numerous domains, RL has shown remarkable promise, serving as a potent instrument for addressing intricate challenges and enhancing autonomous decision-making systems.

## Reinforcement learning based hyper-heuristics

HLS constitutes a vital component of HH, exploring the space of heuristic methodologies and carefully selects the most appropriate LLH, which greatly impacts the overall performance of HH. Therefore, how to efficiently search the space of heuristic algorithms and select the most suitable LLH is of vital significance. RL is a decision-making tool whose tasks are typically described using Markov Decision Process (MDP), which includes the environment state set, the agent action set, the state transition probability function denoted as $P$ and the reward function. Initially, MDP first learns a policy function $\pi$ to map the state to the action that should be taken. In the current state, the agent selects the appropriate action to act on the environment according to the policy $\pi$ and receives the reward from the environment feedback. Then the agent moves to the next state based on the transition probability $P$ (*Liu, Gao & Luo, 2019*; *AlMahamid & Grolinger, 2021*). RL can be used as a vuable method within HLS to meet the requirements of learning and selecting LLH. The execution of LLH is regarded as an action, the improvement of the solution after executing

LLH is perceived as a state. Appropriate LLH is then intelligently selected at different stages of the optimization process. HLS utilizes RL to empower HH in assimilating knowledge and expertise acquired from solved specific problem instances, which is then utilized to tackle unanticipated challenges with greater efficacy (_Udomkasemsub, Sirinaovakul & Achalakul, 2023_).

According to the learning goal, reinforcement learning based hyper-heuristics (RL-HH) can be divided into two categories: value-based reinforcement learning hyper-heuristics (VRL-HH) and policy-based reinforcement learning hyper-heuristics (PRL-HH). VRL-HH uses VRL as HLS to indirectly derive the policy by learning the value function. In contrast, HLS in PRL-HH is PRL, which directly optimizes the policy. According to different VRL methods, VRL-HH can be further subdivided into traditional reinforcement learning hyper-heuristics (TRL-HH) and deep reinforcement learning hyper-heuristics (DRL-HH). TRL-HH uses TRL to learn how to choose suitable LLH, while HLS in DRL-HH is DRL that integrates neural networks into TRL. The specific classification is shown in Table 1. Furthermore, Fig. 4 illustrates the frameworks of the three RL-HHs, which share identical structures, only differing in the HLS component. The top is the configuration of TRL-HH, which employing various score update rules to compute the score of each LLH and choose the most suitable one from LLHs. The middle part shows the framework of DRL-HH, which uses neural network as a method to calculate the score of each LLH and select the appropriate LLH based on these scores. PRL-HH is depicted at the bottom of the figure. The probability of each LLH being selected is obtained through policy network learning, followed by randomly selecting an LLH through sampling.

## VALUE-BASED REINFORCEMENT LEARNING HYPER-HEURISTICS

In VRL, decisions hinge upon either the value of the state value function or the value of the action value function. Then the agent chooses the action corresponding to the state with the value among the next states that may be reached starting from the current state. Regarding VRL as HLS, the specific implementation process involves the following steps: During the initialization phase, RL allocates the identical initial score to each LLH. Each time LLH is invoked, VRL is used to update the score of associated LLH to undergo corresponding dynamic alterations. Depending on the improvement made by the LLH on the current solution, the agent uses value-based methods to update the score of the LLH. In each iteration of the search process, LLHs that lead to an improved solution provide a positive reward to the agent. This is achieved by increasing the score to reward LLHs that enhance the solution. On the contrary, the reward returned by the LLHs that cause the solution to deteriorate is negative to the agent. Penalize poorly performing LLHs by reducing their score. The probability of being selected at each decision point is adaptively adjusted according to the score of each LLH. Always choose the most suitable LLH at each stage of the iteration (_Pylyavskyy, Kheiri & Ahmed, 2020_). The process is shown in Fig. 5.

In VRL-HH, score update rules and neural network usually use the $Q$ function to update the score of LLH. The state action value function $Q(s, LLH_i)$ refers to the cumulative return

**Table 1  Specific classification of RL-HH.**

| Classification | | Main RL method | Representative papers |
|---|---|---|---|
| VRL-HH | TRL-HH | Value update | *Pylyavskyy, Kheiri & Ahmed (2020), Özcan et al. (2010), Nareyek (2003), Sin (2011), Di Gaspero & Urli (2012), Kumari, Srinivas & Gupta (2013), Elhag & Özcan (2018), Lamghari & Dimitrakopoulos (2020), Santiago Júnior, Özcan & Carvalho (2020), Ahmed et al. (2021)* |
| | | Transition probability matrix update | *McClymont & Keedwell (2011), Kheiri & Keedwell (2015), Li, Ozcan & John (2019), Li et al. (2023)* |
| | | Bandit-based update | *Ferreira, Goncalves & Trinidad Ramirez Pozo (2015), Sabar et al. (2015), Zhang et al. (2023a)* |
| | | Q table update | *Choong, Wong & Lim (2018), Watkins & Dayan (1992), Falcão, Madureira & Pereira (2015), Smith et al. (2017), Yao, Peng & Xiao (2018), Ahmed et al. (2020), Mosadegh, Fatemi Ghomi & Süer (2020), Gölcük & Ozsoydan (2021), Zhang & Tang (2021), Cheng et al. (2022), Kanagasabai (2022), Lin, Li & Song (2022), Dantas & Pozo (2022), Ji et al. (2023), Liu et al. (2023b), Zhao, Di & Wang (2023), Zhang et al. (2023b), Zhang et al. (2023c), Zhao et al. (2023), Ozsoydan & Gölcük (2023)* |
| | | Other methods | *Garrido & Castro (2012), Heger & Voss (2021), Lassouaoui, Boughaci & Benhamou (2020), Kemmar, Bouamrane & Gelareh (2021), Cao et al. (2022), Ozsoydan & Gölcük (2022), Yin et al. (2023)* |
| | DRL-HH | Deep Q-network | *Dantas, Rego & Pozo (2021)* |
| | | Double deep Q-network | *Zhang et al. (2022)* |
| | | Dueling double deep Q-network | *Tu et al. (2023)* |
| PRL-HH | | Proximal policy optimization | *Udomkasemsub, Sirinaovakul & Achalakul (2023), Kallestad et al. (2023), Cui et al. (2024)* |
| | | Distributed proximal policy optimization | *Qin et al. (2021)* |

that can be obtained by executing $LLH_i$ in the current state $s$. The definition is shown in Eq. (3).

$$Q(s, LLH_i) = E[R_t | s_t = s, LLH = LLH_i]. \tag{3}$$

In value-based reinforcement learning, the $Q$-value function is generally solved through iterative Bellman equation updates, as shown in Eq. (4).

$$Q_{i+1}(s, LLH_i) = (E_{s' \sim s}[r + \gamma \max_{LLH_{i+1}} Q_i(s', LLH_{i+1}) | s, LLH_i] \tag{4}$$

when $i \rightarrow \infty$, $Q_i$ eventually tends to the optimal, that is, the $Q$-value function finally converges through continuous iteration, thereby obtaining the optimal strategy: $\pi^* = argmax_{LLH_i \in LLH} Q^*(s, LLH_i)$. For VRL-HH, the optimal policy is the LLH selected to be applied.

According to different solution methods of value functions, VRL-HH can be further divided into traditional reinforcement learning hyper-heuristics (TRL-HH) and deep reinforcement learning hyper-heuristics (DRL-HH).

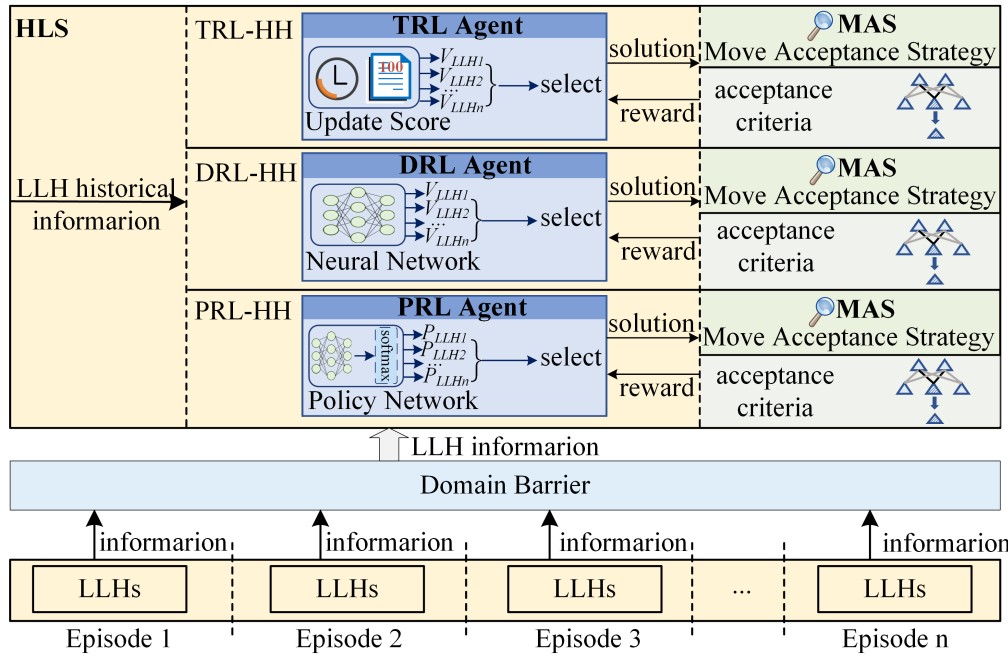

**Figure 4** The framework of RL-HHs.

## Traditional reinforcement learning hyper-heuristics

The TRL-HH typically employs various methods such as value update, transition probability matrix update, bandits-based update, Q table update and other methods to adaptively select the most appropriate LLH.

The concept behind value update involves assigning a certain value to each LLH. The form of value encompasses various forms such as weight, utility value and score. *Nareyek (2003)* proposed a hyper-heuristic evolutionary algorithm based on weight adaptation to learn how to select promising LLH during the search process. *Özcan et al. (2010)* assigned a utility value to each LLH obtained through a predetermined reward and punishment scheme, then selected the appropriate LLH based on the maximum utility value to solve the exam scheduling problem. *Sin (2011)* used the P:1-N:1 strategy to control the weight value of LLH and set upper and lower limits of the weight value, which provides a solution to the exam scheduling problem. *Di Gaspero & Urli (2012)* developed a method based on RL to automatically select LLHs in different problem areas based on value of each LLH. *Kumari, Srinivas & Gupta (2013)* proposed a fast multi-objective hyper-heuristic genetic algorithm MHypGA, which selects LLH based on adaptive weights that change as the search proceeds. *Elhag & Özcan (2018)* extended the grouped hyper-heuristic framework applied to graph coloring, using RL as a heuristic selection method to maintain a utility score for each LLH. *Lamghari & Dimitrakopoulos (2020)* combined RL and tabu search with HH and selected the heuristic based on the score of LLH and the tabu status. *Santiago Júnior, Özcan & Carvalho (2020)* proposed two multi-objective optimization hyper-heuristic algorithms HRISE_M and HRISE_R based on the HRISE framework which select the next LLH to

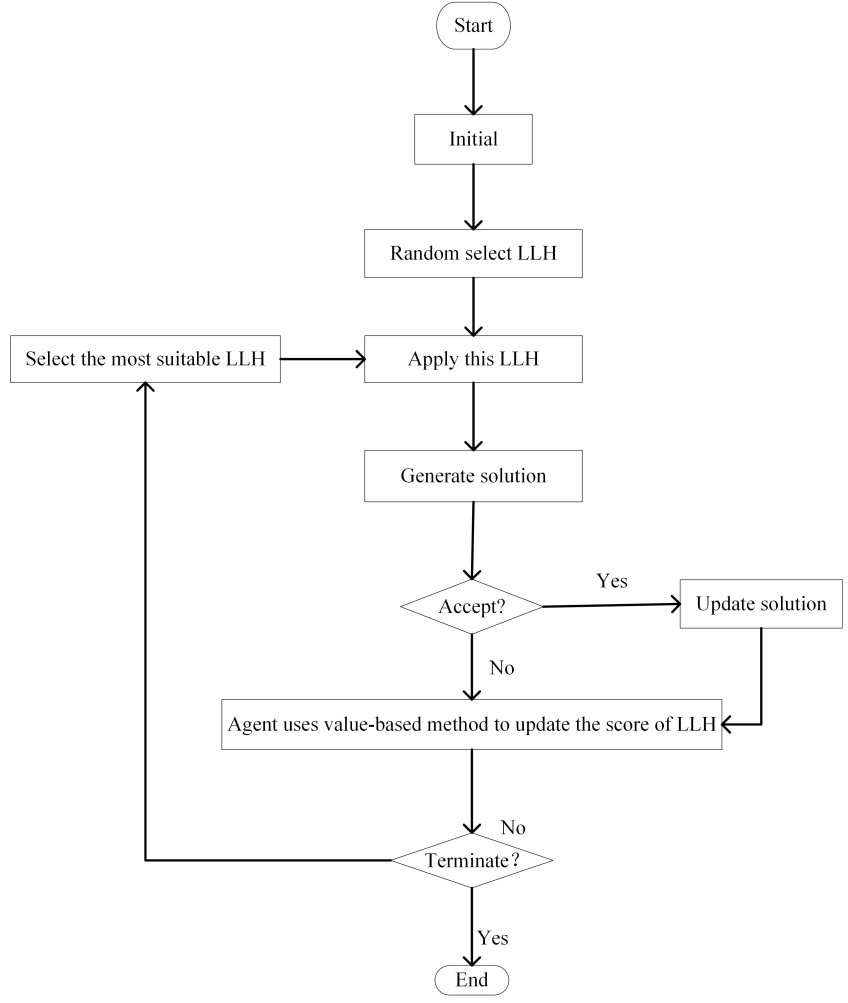

**Figure 5** Flow chart of VRL-HH.

be applied based on the weight value of LLH through the roulette method. *Pylyavskyy, Kheiri & Ahmed (2020)* assign a score to each LLH and change the score according to the improvement plan. RL selects the LLH with the highest score at each stage to optimize the flight connection problem. *Ahmed et al. (2021)* assigned the same initial score to all LLHs, selected LLHs by their scores, and assigned new scores to LLHs according to the performance of the solution in each iteration.

The transition probability matrix explicitly provides the probability of transitioning from one LLH to another. *McClymont & Keedwell (2011)* introduced a new online selection hyper-heuristic algorithm MCHH, which they applied to multi-objective continuous problems. They use Markov chain to simulate the transition probability between LLHs, use RL to update the weight of the transition and adaptively adjust the selection of LLH according to these weights. *Kheiri & Keedwell (2015)* proposed a method based on hidden Markov models to analyze and construct longer low-level heuristic sequences. They

determined the parameters and acceptance strategies of each LLH through the transition probability matrix and the emission probability matrix. *Li, Ozcan & John (2019)* developed a new selection hyper-heuristic framework based on learning automata and implemented two variants HH-LA and HH-RILA. They used learning automata to maintain the transition matrix and select the LLH in the next iteration based on this matrix. In 2023, *Li et al. (2023)* improved on the basis of HH-RILA and proposed HH-mRILA, which utilized a more general initialization strategy and dynamically adjusted parameter settings according to the parameters of the problem being solved.

Bandits can also play the role of HLS as a value-based reinforcement learning method. *Ferreira, Goncalves & Trinidad Ramirez Pozo (2015)* proposed a deterministic selection mechanism based on Multi-Armed Bandits (MAB). They using MAB to select and evaluate the performance of LLH, while using a fixed-size sliding window structure to store reward values for all LLHs, using an upper bound confidence policy to select the next applied LLH. *Sabar et al. (2015)* employed dynamic MAB extreme value rewards as an online selection mechanism, which determine the most appropriate LLH for the next iteration by detecting the average reward change of the current best LLH to solve combinatorial optimization problems. *Zhang et al. (2023a)* proposed a new selective hyper-heuristic algorithm adaptive bandit-based selection hyper-heuristic (HH-AB), which uses the bandit-based method as a learning mechanism to update the quality of each LLH. By iteratively learning, selecting, and applying LLHs, HH-AB addresses multi-objective optimization problems.

There are many other traditional reinforcement learning methods used as HLS. *Garrido & Castro (2012)* proposed an improved adaptive hyper-heuristic algorithm, based on simple RL to assign reward or penalty values to LLH and guide the selection of LLH. *Heger & Voss (2021)* used the RL algorithm to select LLH based on the sorting rules, proving the ability of RL to dynamically change the sorting rules to reduce the average delay of the system. *Lassouaoui, Boughaci & Benhamou (2020)* used Thompson sampling selection strategy to evaluate the behavior of LLH, updated the learning mechanism based on Beta probability law. *Kemmar, Bouamrane & Gelareh (2021)* utilized RL based on a scoring system to dynamically update the weight value of LLH every time it is called, thereby solving the hub location problem in round-trip service applications. *Cao et al. (2022)* developed a reinforcement learning hyper-heuristic inspired by probabilistic matching and applied it to structural damage identification. *Ozsoydan & Gölcük (2022)* adopted a feedback mechanism as HLS, which promotes more successful algorithms while implicitly hindering the development of the remaining algorithms. *Yin et al. (2023)* proposed a hyper-heuristic algorithm based on reinforcement learning (HHRL) to solve the task scheduling problem in cloud computing. HHRL uses an advanced heuristic method based on a reward table updated with iterations.

With the emergence of Q-Learning (*Watkins & Dayan, 1992*), a large number of HH have arisen that use Q table to record state action value functions and select LLH based on this function value. These types of HH have successfully addressed combinatorial optimization problems such as scheduling and allocation. For the workshop scheduling problem, *Zhang & Tang (2021)* embedded Q-Learning into the HH framework, introducing Q-Learning-based hyper-heuristics (QHH), *Cheng et al. (2022)* devised a multi-objective

Q-learning-based hyper-heuristic with Bi-criteria selection (QHH-BS), *Zhao, Di & Wang (2023)* proposed a hyper-heuristic with Q-Learning (HHQL). Additionally, *Zhang et al. (2023b)* proposed a Q-Learning-based hyper-heuristic evolutionary algorithm (QLHHEA), *Zhang et al. (2023c)* designed a Q-Learning-based hyper-heuristic evolutionary algorithm (QHHEA). All these algorithms apply Q-Learning as HLS, manipulating the choice of LLH at each decision point based on feedback information from different stages, effectively addressing the workshop scheduling problem. *Falcão, Madureira & Pereira (2015)* utilized Q-Learning to independently select the LLH and corresponding parameters used in the optimization process to solve scheduling problems in the manufacturing system. *Lin, Li & Song (2022)* presented a Q-Learning based hyper-heuristic (QHH) to solve the semiconductor final testing scheduling problem with the maximum time span. For allocation problems, *Ji et al. (2023)* suggested a novel Q-Learning-based hyper-heuristic evolutionary algorithm (QLHHEA), which handles the task allocation problem in a self-learning manner. *Liu et al. (2023b)* proposed a parallel HH based on Q-Learning (QL-PHH), which considers the combination states and utilizes Q-learning to select suitable LLH, aiming to quickly solve corridor allocation problems. The use of Q-learning-based HH has led to the effective resolutions of numerous real-world problems as well. *Mosadegh, Fatemi Ghomi & Süer (2020)* utilized Q-learning as the RL to develop a novel hyper simulated annealing (HSA), which was employed to investigate real-world mixed-model sequencing problems. *Kanagasabai (2022)* proposed an algorithm using Q-Learning and hyper-heuristic (QH) for the power loss reduction problem. *Zhao et al. (2023)* introduced a selection hyper-heuristic framework with Q-learning (QLSHH), validating the effectiveness of the algorithm on a common engineering problem of pressure vessel design. Fehmi et al. *Ozsoydan & Gölcük (2023)* proposed using Q-Learning as an LLH recommendation system to analyze the performance of the optimizers used, which provided a solution to the set-union knapsack problem.

All TRL-HH implementations mentioned above employ conventional VRL as the learning strategy of HH. The approach calculates the score of LLH according to the information of LLH and different score update rules. The most suitable LLH is then selected and applied to generate candidate solutions. MAS evaluates the acceptance of the candidate solution and provides feedback to HLS at the same time, gradually improving the performance of HLS in selecting LLH. The framework of TRL-HH is shown at the top of Fig. 4.

## Deep reinforcement learning hyper-heuristics

TRL-HH, which uses methods such as transition probability matrix update, Q table update as HLS, is only suitable for problems with small action spaces and sample spaces. These TRL methods lack scalability and are limited to low-dimensional problems (*Arulkumaran et al., 2017*). Nevertheless, complex tasks frequently involve large state spaces and continuous action spaces. TRL methods struggle to handle such complex tasks with high-dimensional features. In order to address this challenge, RL is combined with deep learning (DL), giving rise to the concept of DRL. RL excels in decision-making but is helpless on perception problems. while although lacking certain decision-making capability, DL demonstrates

robust perceptual ability. DL can effectively classify the environment and improve the performance of RL (*Qin et al., 2021*). DRL synergistically harnesses the strengths of both RL and DL, integrating neural network into RL to address the challenges associated with perception and decision-making in complex systems. In DRL, the deep neural network possesses the capability to autonomously discern and abstract sophisticated features directly from raw data inputs (*Zhang et al., 2022*). DRL implements end-to-end learning, acquiring knowledge directly from original input to output actions, which simplifies the design process and enhances scalability. With continuous learning and adjustment, the deep neural network gradually refines. DRL dynamically adjusts to evolving environments and showcasing superior generalization capabilities. DRL is an artificial intelligence method that is closer to the way humans think, offering advantages in addressing large-scale and high-dimensional problems. Treat DRL as HLS of HH, DRL-HH is proposed. The DRL agent uses deep neural network to approximate the reward value function, ultimately selecting the most appropriate LLH based on the obtained value. The framework of DRL-HH is shown in the middle part of Fig. 4. DRL-HH can improve the performance and robustness of HH, enabling superior handling of high-dimensional data.

Limited study has been conducted on DRL-HH. *Dantas, Rego & Pozo (2021)* investigated a selection hyper-heuristic which uses a DQN. This modeled the task of selecting LLH as a Markov decision process, used DQN to select LLH based on the current observed state representation, and iteratively improved solutions to the vehicle routing problem (VRP) and the traveling salesman problem (TSP). *Zhang et al. (2022)* used DDQN to train the selection module of HH, which provides better ability to process high-dimensional data. It also adopts an experience replay strategy to handle uncertainty issues more effectively. *Tu et al. (2023)* proposed a new DRL-HH that combines DRL's end-to-end sequential decision-making capability with a selection hyper-heuristic. D3QN is used as an advanced heuristic reinforcement learning algorithm to collect decisions, status and rewards from the problem environment to improve its own performance. It also uses a feature fusion method to extract key features of the online environment to solve the online packaging problem.

Whether it is TRL-HH or DRL-HH, the optimal strategies in these VRL-HHs are indirectly obtained by the agent using the argmax function to calculate the Q value of LLH. Although DRL makes up for the shortcomings of TRL, it can only solve the problem of high-dimensional state spaces and cannot solve the problem of high-dimensional action spaces. These VRL-HHs might exhibit drawbacks such as a limited description of the problem and a random nature of the policy, which are commonly encountered in the search space of LLHs (*Qin et al., 2021*).

## POLICY-BASED REINFORCEMENT LEARNING HYPER-HEURISTICS

PRL exhibits greater adaptability when confronted with problems presented in high-dimensional and continuous action spaces. By directly learning the optimal policy, PRL makes up for the shortcomings of VRL, thereby enhancing stability. Therefore, PRL is used

to solve the problems associated with high-dimensional action spaces and serves as HLS to propose PRL-HH to further improve algorithm performance. The framework of PRL-HH is shown in the bottle of Fig. 4.

PRL usually parameterizes the policy and constructs a policy model. Policy function in the policy network is typically represented as $\pi$ $(s,\ LLH_i,\ \theta)$, which indicating the probability of directly outputting $LLH_i$ in state $s$. The model is controlled by $\theta$, we can find the optimal strategy by finding the most appropriate rparameter $\theta$. Establishing an objective function is necessary to find the optimal strategy $\pi$, and then $\theta$ is found through various extreme value methods. Iteratively update the parameters $\theta$ of the policy function to maximize the expected return leads to convergence to the optimal policy. The update formula of parameter $\theta$ is:

$$\theta = \theta + \alpha \nabla J(\theta) \tag{5}$$

where $\theta$ is the parameter of the policy gradient, $\alpha$ is the learning rate, and is the gradient of the expected reward.

Proximal policy optimization (PPO) stands as a popular PRL method that is used as HLS to select LLH. *Kallestad et al. (2023)* proposed deep reinforcement learning hyper-heuristic (DRLH), which uses PPO to train a DRL agent. This agent effectively utilizes search state information in each iteration to make better decisions in selecting the next LLH to be applied. DRLH operates at a micro-level in order to adapt to different problem conditions and settings. Moreover, the performance of DRLH is not negatively affected by an increase in the number of available LLHs. *Udomkasemsub, Sirinaovakul & Achalakul (2023)* proposed a new policy-based hyper-heuristic framework (PHH) whose uses PPO as learning algorithm to represent the probability distribution of available LLHs under a given environmental state. PHH also contains an experience buffer, which stores experience samples used for training and improving the policy. PHH is applied to solve workflow scheduling problems in hybrid cloud. *Cui et al. (2024)* proposed a new DRL HH framework to solve real-world multi-period portfolio optimization problems. The heuristic selection module is trained using the PPO algorithm, which computes an update at each step to optimize the cost function, and ultimately selects the LLH based on the two state vectors and the experience of the DRL agent.

Distributed proximal policy optimization (DPPO) serves as a distributed implementation of PPO, which is capable of multi-thread parallelization. *Qin et al. (2021)* developed a novel reinforcement learning-based hyper-heuristic (RLHH), whose HLS uses a distributed proximal policy optimization (DPPO) to select the appropriate LLH at each decision point. RLHH utilizes a multi-threading method based on asynchronous advantage actor-critic (A3C) to accelerate the entire training process and converge the policy gradient faster. And then uses DPPO to calculate the policy gradient to find the next LLH to be executed. This algorithm effectively overcomes the possible limitations of VRL and effectively solves the heterogeneous vehicle routing problem.

PRL often requires policy optimization, which can involve highly complex numerical optimization problems. In many cases, policy-based approaches frequently incur substantial time overheads, particularly in scenarios featuring high-dimensional state spaces and action

spaces. Moreover, policy-based methods explore in the policy space and require more samples to learn effective policies, making them inferior to value-based methods in terms of sample efficiency. Additionally, PRL is also more susceptible to training instabilities, which can make the training process extremely difficult and make it difficult for the algorithm to converge. Due to these challenges, there is currently little limited on PRL-HH.

## APPLICATION

As a powerful optimization method, RL-HH has very significant practical value. RL-HH leverages the adaptive capabilities of RL to learn and refine strategies over time, alongside the flexibility of HH to select and combine LLHs effectively. Such a combination allows RL-HH to tackle diverse challenges and presents a robust approach to solving complex optimization problems. By enabling a more nuanced exploration of the solution space and iterative learning from interactions, RL-HH has demonstrated exceptional value in areas where traditional methods fall short, including resource allocation, scheduling and automated decision-making. At present, RL-HH has been successfully implemented in these optimization challenges, which demonstrates the effectiveness of RL-HH. Table 2 shows the application areas that RL-HH has implemented.

In addition to the application areas mentioned in Table 2, RL-HH can also be deployed in many other scenarios. For example, RL-HH can be used in the medical and health field to assist doctors in formulating more precise treatment plans to improve the quality and efficiency of medical services. In the field of energy management, RL-HH can also be applied to help operators intelligently schedule energy to achieve energy conservation and emission reduction. RL-HH has strong practicality. In the future, we need to further explore RL-HH. Apply RL-HH to solve more real-life problems and develop broader application prospects.

## DISCUSSION

VRL-HH primarily revolves around estimating the value of each possible LLH in the current iteration, This method focuses on learning the value function. The core mechanism is to assign a value to all LLHs during initialization. At each decision point, the most appropriate LLH is chosen based on the value or through a policy derived from the value function. After taking an LLH and getting the reward, the value of the previous state-LLH pair is updated iteratively using the Bellman equation. Through the learning process, the algorithm will always choose the most suitable LLH at the moment.

Unlike VRL-HH, PRL-HH is characterized by directly optimizing the policy that maps state to LLH without using value functions as an intermediary. The key to PRL-HH is to parameterize the policy, this policy defines the probability of selecting each LLH in the current state. Then LLH is selected based on probabilities derived from the policy. The policy parameters are adjusted based on the gradient of the expected reward. So that the parameters are moved in the direction that maximally increases the expected reward. As a result, PRL-HH gradually improves the mechanism for selecting LLH.

**Table 2  Real-life applications of RL-HH.**

| Application areas | Representative papers |
|---|---|
| Scheduling problem | *Özcan et al. (2010), Ferreira, Goncalves & Trinidad Ramirez Pozo (2015), Sin (2011), Lamghari & Dimitrakopoulos (2020), Yin et al. (2023),* |
| | *Falcão, Madureira & Pereira (2015),Zhang & Tang (2021), Cheng et al. (2022), Lin, Li & Song (2022), Zhao, Di & Wang (2023),* |
| | *Zhang et al. (2023b), Zhang et al. (2023c)* |
| Allocation Problem | *Ahmed et al. (2021), Ji et al. (2023), Ji et al. (2023)* |
| Traveling salesman problem | *Pylyavskyy, Kheiri & Ahmed (2020), Udomkasemsub, Sirinaovakul & Achalakul (2023), Mosadegh, Fatemi Ghomi & Süer (2020)* |
| Vehicle routing problem | *Garrido & Castro (2012), Sabar et al. (2015), Yao, Peng & Xiao (2018), Mosadegh, Fatemi Ghomi & Süer (2020), Zhang & Tang (2021)* |
| Packing problem | *Ferreira, Goncalves & Trinidad Ramirez Pozo (2015), Ahmed et al. (2020), Gölcük & Ozsoydan (2021)* |
| Knapsack problem | *Zhang et al. (2023a), , Gölcük & Ozsoydan (2021)* |
| Vehicle crashworthiness problem | *Santiago Júnior, Özcan & Carvalho (2020), Li, Ozcan & John (2019), Zhang et al. (2023a)* |
| Multi-period portfolio optimization problem | *Cheng et al. (2022)* |
| Structural damage identification problem | *Cao et al. (2022)* |

The difference between VRL-HH and PRL-HH is mainly reflected in the following points. First, in terms of complexity and efficiency, VRL-HH can be computationally efficient in discrete and smaller action spaces, whereas PRL-HH can handle complex, continuous action spaces more effectively. Furthermore, as far as applicability is concerned, VRL-HH is often preferred in environments where an explicit value can be assigned to LLHs, while PRL-HH excels in scenarios where modeling the environment directly through policies can offer better performance without the intermediate step of value estimation.

Both VRL-HH and PRL-HH have had a profound impact on theory and practice, and each has unique advantages and limitations.

From a theoretical perspective, VRL-HH is able to handle problems in continuous state and action spaces. Because of the value-based nature, VRL-HH can provide deep insights into the problem structure. However, VRL-HH may also face the challenge of the curse of dimensionality, especially when the state or action space is very large. Furthermore, for some complex problems, it may be difficult to accurately estimate the value. PRL-HH performs better on problems with high dimensions and continuous action spaces. Due to the property of directly optimizing strategy, PRL-HH can usually converge to a better strategy faster. But PRL-HH also faces the problem of policy degradation. Moreover, PRL-HH generally requires more computing resources because it requires direct optimization of the parameters of the policy.

From the perspective of the impact on practice, VRL-HH accurately evaluates the value function and is suitable for scenarios that require accurate evaluation of future potential

returns, such as resource allocation problems. However, for complex problems with high real-time requirements or the need to process large amounts of data, VRL-HH requires a large amount of storage and computing resources, which is impractical in real-time or resource-constrained environments. In comparison, PRL-HH is more suitable for solving complex problems. PRL-HH can usually better adapt to environmental changes and is suitable for solving real-time challenges. However, PRL-HH requires more interactive samples to learn effective strategies, which is a limiting factor in practical applications where sample acquisition costs are high.

Although a lot of progress has been made in RL-HH, many limitations in this field still exist as follows:

(1) Lack of theoretical knowledge analysis of RL-HH. The theoretical foundation of RL-HH necessitates further exploration, particularly in terms of theoretical analysis encompassing the RL-HH's convergence, convergence speed, and stability.

(2) One of the fundamental challenges in RL-HH is the design of RL. The efficacy of RL-HH mainly relies on the capability of RL. Improperly designed RL can lead to poor overall performance of RL-HH.

(3) The lack of adaptive research is also an obstacle to the current development of RL-HH. The existing RL-HH cannot adapt well to the characteristics and needs of different problem areas.

(4) The computational cost of RL-HH is also one of the unresolved limitations. This limitation is a critical concern when aiming for real-time applications or scenarios where computational resources are limited.

(5) Both VRL-HH and PRL-HH have their own limitations. How to combine them to create a more powerful and flexible RL-HH has yet to be solved.

In view of the above limitations, there are many future research courses on RL-HH.

(1) Adequate theoretical analysis is crucial for the development of RL-HH. A comprehensive analysis of these theoretical aspects will not only enhance our understanding of RL-HH but also contribute to the robustness and reliability of RL-HH in various problem-solving scenarios. Theoretical advancements can help us better understand RL-HH, thereby fostering the applicability and effectiveness of RL-HH in different fields.

(2) The performance of RL directly affects the effect of RL-HH. In the future we can integrate advanced and efficient RL algorithms into the framework of HH. The learning and decision-making capabilities of HH can be significantly enhanced by refining and incorporating state-of-the-art RL.

(3) Developing adaptive RL-HH is also a focus of future research efforts. In-depth research on adaptive RL-HH to dynamically adapt to different properties and need of various problem domains is urgently needed. The adaptability of RL-HH to environmental changes can be enhanced through research on multi-task learning, transfer learning and combination with model predictive control.

(4) Reducing the computational cost of RL-HH is also a course worthy of research in the future. The reduction in computing costs is beneficial to speeding up the execution of RL-HH and can improve the overall computing resource utilization. In future research, computing costs can be reduced by developing more efficient models such as using

approximate dynamic programming or using caching technology to reduce redundant calculations.

(5) Combining the respective advantages of VRL-HH and PRL-HH to develop new hybrid RL-HH is also a future direction worth exploring. Hybrid RL-HH can take full advantage of the stability and accuracy of VRL-HH and the flexibility of PRL-HH in processing high-dimensional space.

At the same time, with the continuous advancement of related technologies and the emergence of new problems, we also need to continue to explore and develop new RL-HH to deal with various challenges.

## CONCLUSION

RL-HH is a frontier of multi-field cross-research. As an important optimization technology, RL-HH has received widespread attention and application. By integrating reinforcement learning with hyper-heuristic evolutionary algorithm, RL-HH endeavors to improve the efficiency and performance of hyper-heuristic evolutionary algorithm. Research in this burgeoning field has made significant progress. The problem-solving ability of RL-HH has been verified in many literatures, with grate potential for practical application. In this research, we elucidate the fusion of value-based and policy-based RL methods with HH, delineating the distinct advantages offered by various RL-HHs. Furthermore, we provide a comprehensive summary of existing RL-HH methodologies. Through this study, we hope to provide researchers in related fields with a comprehensive understanding of RL-HH, thereby promoting further development and research on RL-HH.

### Funding
This work was supported by the National Key Technologies Research and Development Program (2020YFB1712401, 2018YFB1701400), the Key Special Technologies Research and Development Program in Henan Province (231111211900), the Major Science and Technology Project in Henan Province (201300210500), the Key Scientific Research Project of Colleges and Universities in Henan Province (23A520015), and the Henan Provincial Science and Technology Research Project (232102210090). There was no additional external funding received for this study. The funders had no role in study design, data collection and analysis, decision to publish, or preparation of the manuscript.

### Grant Disclosures
The following grant information was disclosed by the authors:
The National Key Technologies Research and Development Program: 2020YFB1712401, 2018YFB1701400.
Key Special Technologies Research and Development Program in HenanProvince: 231111211900.
Major Science and Technology Project in Henan Province: 201300210500.

Key Scientific Research Project of Colleges and Universities in Henan Province: 23A520015. Henan Provincial Science and Technology Research Project: 232102210090.

## Competing Interests

The authors declare there are no competing interests.

## Author Contributions

- Cuixia Li conceived and designed the experiments, authored or reviewed drafts of the article, and approved the final draft.
- Xiang Wei conceived and designed the experiments, performed the experiments, analyzed the data, prepared figures and/or tables, and approved the final draft.
- Jing Wang performed the experiments, analyzed the data, prepared figures and/or tables, and approved the final draft.
- Shuozhe Wang performed the experiments, analyzed the data, prepared figures and/or tables, and approved the final draft.
- Shuyan Zhang conceived and designed the experiments, authored or reviewed drafts of the article, and approved the final draft.

## Data Availability

This is a literature review.

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
