# Peer review of "A review of reinforcement learning based hyper-heuristics"

_PeerJ Computer Science, doi:10.7717/peerj-cs.2141_

## Round 0.1 · original submission · Major Revisions

Dear authors,

Thank you for submitting your literature review article. Reviewers have now commented on your study. Your article has not been recommended for publication in its current form. However, we encourage you to address the concerns and criticisms of the reviewers and to resubmit your article once you have updated it accordingly. Furthermore, adding a comprehensive discussion for synthesis of findings, implications, future research, and limitations will be better.

Reviewer 2 has asked you to provide specific references. You are welcome to add them if you think they are relevant. However, you are not obliged to include these citations, and if you do not, it will not affect my decision.

Best wishes,

**Language Note:** PeerJ staff have identified that the English language needs to be improved. When you prepare your next revision, please either (i) have a colleague who is proficient in English and familiar with the subject matter review your manuscript, or (ii) contact a professional editing service to review your manuscript. PeerJ can provide language editing services - you can contact us at [email protected] for pricing (be sure to provide your manuscript number and title). – PeerJ Staff

Reviewer 1 ·

Basic reporting

The manuscript is written in clear and professional English. The manuscript offers a comprehensive introduction to the field of RL-HH. The authors committed to the journal's format and scientific research structure. The manuscript falls within the scope of the journal. The authors acknowledged the existence of HH studies but utilizing RL in HH offers a recent and demanding perspective in the rapidly evolving area. The authors succeeded in providing an effective introduction based on the motivations for and advancements in RL-HH. Given the nature of the manuscript, it focuses on summarizing and categorizing existing research.

Experimental design

The manuscript provides a comprehensive review of RL-HH, emphasizing technical and methodological detail that met to academic standards. The manuscript shows a systematic literature search method for thorough and unbiased coverage of RL-HH, ensuring the possibility of replication. The manuscript from introduction through to conclusion, alongside appropriate source citation and use of illustrative figures and tables, contributes to a reasonable presented overview of RL-HH current state and advancements.

Validity of the findings

The manuscript addresses the impact and novelty of RL-HH by detailing the integration of reinforcement learning with hyper-heuristics, suggesting it fills a significant knowledge gap. Conclusions related directly to the research question, summarizing state and advancements in RL-HH concisely. It builds a strong argument based on systematic categorization and analysis of RL-HH, aligned with initial goals for a comprehensive understanding of the field.

Additional comments

Incorporating a comprehensive discussion section into the manuscript on RL-HH could substantially enrich the work. This section would serve to merge insights from value-based and policy-based RL-HH studies, emphasizing key themes and differences, and critically analyzing the implications for theory and practice. It would also offer a more nuanced look at future research course and critically assess the field's current limitations.

Cite this review as

Reviewer 2 ·

Basic reporting

The paper is well-written and well-organized.

Experimental design

I believe that this research can be extended by including real-life applications.
Therefore, an additinal section that is devoted to real-life applications can be opened.

Validity of the findings

There is not a clear distinction between HHs and RL. In many papers these are used as interchangable terms. Maybe the authors should mention about this.

Additional comments

Before diving into details, some fundamentals of RL algorithms should be provided. For example, Q learning and Bellman eq should be given at earlier sections. Moreover, the differencens between SARSA and Q learning should be clarified. Pseudo codes should be provided.
I beleive that there are some missing studies that should take place.

-Q-learning and hyper-heuristic based algorithm recommendation for changing environments
-Artificial search agents with cognitive intelligence for binary optimization problems
-A reinforcement learning based computational intelligence approach for binary optimization problems: The case of the set-union knapsack problem
-Iterated greedy algorithms enhanced by hyper-heuristic based learning for hybrid flexible flowshop scheduling problem with sequence dependent setup times: a case study...
-Evolutionary and population-based methods versus constructive search strategies in dynamic combinatorial optimization

Cite this review as

---

## Round 0.2 · accepted · Accept

Dear authors,

Thank you for the revision and for clearly addressing all the reviewers' comments. I confirm that the paper is improved. Your paper is now acceptable for publication in light of this revision.

Best wishes,

Reviewer 1 ·

Basic reporting

The manuscript is well-written.

Experimental design

The manuscript provides a comprehensive review of RL-HH, emphasizing technical and methodological detail that met to academic standards.

Validity of the findings

The discussion added.

Cite this review as

Reviewer 2 ·

Basic reporting

The current revision is well-written and well-organized. I have no further questions.

Experimental design

Although it is not perfect I believe that this study can be published with its current form.

Validity of the findings

I have no further suggestions regarding the validity of findings.

Additional comments

Congrats

Cite this review as